# Inhibition Effects and Mechanisms of Marine Polysaccharide PSSD against Herpes Simplex Virus Type 2

**DOI:** 10.3390/md21060364

**Published:** 2023-06-18

**Authors:** Han Yan, Jie Wang, Jiayi Yang, Zhongqiu Xu, Chunxia Li, Cui Hao, Shixin Wang, Wei Wang

**Affiliations:** 1Key Laboratory of Marine Drugs, Ministry of Education, Shandong Provincial Key Laboratory of Glycoscience and Glycoengineering, School of Medicine and Pharmacy, Ocean University of China, 5 Yushan Road, Qingdao 266003, China; yanhan7352@163.com (H.Y.); 17860765781@163.com (J.W.); yangjy904@163.com (J.Y.); xzq1456958571@163.com (Z.X.); lchunxia@ouc.edu.cn (C.L.); 2Medical Research Center, Affiliated Hospital of Qingdao University, Qingdao 266000, China; 3The Laboratory of Marine Glycodrug Research and Development, Marine Biomedical Research Institute of Qingdao, Qingdao 266100, China

**Keywords:** herpes simplex virus, marine polysaccharide, inhibition effects, molecular mechanism, membrane fusion, genital herpes

## Abstract

Genital herpes is a common sexually transmitted disease mainly caused by herpes simplex virus type 2 (HSV-2), which can increase the risk of HIV transmission and is a major health problem in the world. Thus, it is of great significance to develop new anti-HSV-2 drugs with high efficiency and low toxicity. In this study, the anti-HSV-2 activities of PSSD, a marine sulfated polysaccharide, was deeply explored both in vitro and in vivo. The results showed that PSSD had marked anti-HSV-2 activities in vitro with low cytotoxicity. PSSD can directly interact with virus particles to inhibit the adsorption of virus to the cell surface. PSSD may also interact with virus surface glycoproteins to block virus-induced membrane fusion. Importantly, PSSD can significantly attenuate the symptoms of genital herpes and weight loss in mice after gel smear treatment, as well as reducing the titer of virus shedding in the reproductive tract of mice, superior to the effect of acyclovir. In summary, the marine polysaccharide PSSD possesses anti-HSV-2 effects both in vitro and in vivo, and has potential to be developed into a novel anti-genital herpes agent in the future.

## 1. Introduction

Herpes simplex virus type 1 (HSV-1) and type 2 (HSV-2) are double-stranded DNA viruses belonging to the α subfamily of *Herpesviridae* [1]. Among them, HSV-1 mainly causes cold sores, herpetic encephalitis (HSE), and herpetic keratitis (HSK) [2,3], and HSV-2 mainly causes genital herpes [4]. HSV-1 can also reach the brain and cause herpes encephalitis, and may lead to nervous system diseases such as Alzheimer’s disease (AD) [5,6,7]. Genital herpes caused by HSV-2 can increase the risk of HIV (AIDS) transmission [8], partially through impaired integrity of the vaginal mucosal barrier due to genital lesions [9]. At present, acyclovir and its derivatives, such as valacyclovir, are mainly used to treat HSV by inhibiting viral DNA replication, but long-term use can easily to lead to drug resistance [10,11]. Therefore, it is of great importance to develop novel anti-HSV drugs with high efficiency and low toxicity.

Marine organisms have produced a large number of structurally novel marine active molecules in special environments. Recently, research on the anti-HSV effects of marine algae polysaccharides, marine peptides, marine alkaloids, and terpenoids have been continuously reported [12]. The sulfated polysaccharides derived from brown seaweeds have recently been demonstrated to possess broad inhibitory activities against viruses, such as human immunodeficiency virus (HIV) [13] and herpes simplex virus (HSV) [14]. The alginate-derived polymannuroguluronate sulfate (PMGS) was reported to possess significant inhibition effects against human papillomavirus infection both in vitro and in vivo through direct binding to HPV L1 protein to block the virus entry process [15]. Thus, the sulfated derivatives of alginate have potential to be developed into novel antivirals in the future.

Propylene glycol alginate sodium sulfate (PSS), a sulfated derivative of alginate polysaccharide, has been used for preventing and treating hyperlipidemia and ischemic cardio-cerebrovascular diseases in China for 30 years [16]. Recently, our group graded the alginate polysaccharide PSS and obtained four components with different molecular weights by using gel filtration separation. Through activity screening, we found that the PSS derivative (PSSD) with low molecular weight had marked anti-HSV effects in vitro, thus the anti-HSV-2 activities and mechanisms of PSSD were investigated both in vitro and in vivo in this study. The results indicated that PSSD may directly inactivate viral particles and interact with virus surface glycoproteins to block HSV adsorption and entry processes. Importantly, vaginal gel therapy of PSSD markedly attenuated the symptoms of genital herpes in mice and reduced the virus titers of mouse genital tract exfoliated virus, suggesting that PSSD has potential to be developed into a novel anti-genital herpes agent in the future.

## 2. Results

### 2.1. Structure Characterization of Marine Polysaccharide PSSD

PSSD is a mixture of alginate polysaccharide with different substituents and degrees of polymerization, and the structure of the repeat unit of PSSD is shown in Figure 1a. The average molecular weight of PSSD was about 6.7 kDa determined by HPGPC (Figure 1b), and the sulfate content was about 34.2% determined by the IC method. The structure of PSSD was determined by Fourier transform infrared spectroscopy (FT-IR) and nuclear magnetic resonance (NMR) analysis (Figure 1c,d). As shown in Figure 1c, the FT-IR of PSS and PSSD were similar, indicating that PSSD had not changed the structure of PSS. In Figure 1c, the bands at 3209 cm^−1^ were the stretching vibration peak of O-H; 1620 cm^−1^ and 1450 cm^−1^ were the symmetric and asymmetric stretching vibration peaks of COO-, while 1260 cm^−1^ and 840 cm^−1^ were the symmetric stretching vibration peak of S=O and the asymmetric stretching vibration peak of C-O-S, respectively; 1006 cm^−1^ was the stretching vibration peak of C-O in C-O-H; 937 cm^−1^ was the asymmetric stretching vibration of the pyranose ring; and 881 cm^−1^ was the C-H deformation vibration of a mannuronic acid isomer.

In the ^13^C NMR spectrum of PSSD (Figure 1d), the peaks at 174.62 ppm and 99.68 ppm were attributed to C-6 and C-1 [15], and the signals of C-2, C-3, C-4, and C-5 were packed in a narrow region between 60 and 80 ppm (Figure 1d). Compared with the signals in the corresponding PSS, the peaks of carbohydrate had no changes. The significant signals at 16.68 ppm and 69.26 ppm for PSSD were the –CH3 and –CH2– of propylene glycol, which indicated that the chemical structure of PSS was not destroyed. In summary, the above results from the chemical analysis showed that PSS and PSSD had similar structural features, and mainly differed in their molecular weights [17].

### 2.2. PSSD Suppresses HSV Multiplication In Vitro with Low Toxicity

The cytotoxicity of PSSD in Vero and HeLa cells was firstly evaluated by MTT assay [18]. The results showed that PSSD exhibited no significant cytotoxicity at the concentrations from 6.25 to 800 μg/mL (Figure 2a,b). PSSD showed some cytotoxicity to HeLa cells at 800 μg/mL but without statistical significance (Figure 2b). The CC_50_ (50% cytotoxicity concentration) value for PSSD was more than 1000 μg/mL. These results were used to determine the dose range of PSSD for the subsequent experiments.

PSSD was then assayed for its ability to inhibit HSV multiplication in vitro using a CPE inhibition assay and western blot assay [18]. As shown in Figure 2a,b, PSSD (12.5–200 μg/mL) treatment dose-dependently promoted the viability of HSV infected cells, and the IC_50_ values obtained for PSSD inhibition of HSV-2 in Vero and HeLa cells were about 23.46 ± 2.73 μg/mL and 39.06 ± 4.44 μg/mL, respectively. Aside from that, the results of western blot assay indicated that PSSD (12.5–100 μg/mL) treatment significantly reduced the expression levels of virus ICP27 protein in a concentration-dependent manner (*p* < 0.01) (Figure 2c,d). Therefore, PSSD can suppress HSV-2 multiplication in vitro with low toxicity.

### 2.3. Influence of Different Treatment Conditions of PSSD on HSV-2 Infection

The time-of-addition assay was then performed to explore the stage(s) at which PSSD exerted its inhibition actions as described previously [18,19]. As shown in Figure 3a, the results of the CPE inhibition assay showed that pretreatment of HSV-2 (MOI = 0.1) with PSSD (12.5–50 μg/mL) for 1 h before infection markedly improved the viability of HSV-2-infected cells in a dose-dependent manner, suggesting that PSSD may have direct interaction with HSV particles. Treatment of PSSD during adsorption or after adsorption also possessed significant inhibition on HSV-2 multiplication (Figure 3a). Similarly, the western blot assay indicated that pretreatment of virus with PSSD (100 μg/mL), and treatment of PSSD during adsorption or after adsorption all significantly reduced the expression of ICP27 proteins in HSV-2-infected Vero cells (*p* < 0.01) (Figure 3b,c). Thus, PSSD may interact with virus particles to block adsorption or inhibit some steps after virus adsorption.

Furthermore, another time course study within 8 h was also performed to further determine the viral stage post-adsorption inhibited by PSSD. As shown in Figure 3d,e, treatment with PSSD (100 μg/mL) during the first two hours after adsorption (0–2 h p.i.) significantly reduced the production of ICP27 to about 40% of the virus control group (*p* < 0.01). However, no significant inhibition on ICP27 production was noted when PSSD was added later than 2 h post-infection (2–8 h p.i.) (Figure 3d,e). Thus, PSSD may also inhibit the early steps (0–2 h p.i.) of the HSV life cycle after adsorption.

### 2.4. PSSD Possesses Direct Actions on HSV Particles to Block Virus-Induced Membrane Fusion

Since PSSD may interact with virus particles to block HSV infection, we further explored whether PSSD has direct inactivation effects on HSV particles by performing the plaque reduction assay as described previously [19,20]. As shown in Figure 4a,b, pretreatment of HSV-2 with PSSD (12.5–100 μg/mL) significantly reduced the number of plaques in HSV-2-infected Vero cells in a dose-dependent manner (*p* < 0.01), suggesting that PSSD may be able to inactivate viral particles directly.

The time-of-addition assay indicated that PSSD may block the early infection process of HSV after adsorption, so we then explored whether PSSD could inhibit the virus-induced membrane fusion process. As shown in Figure 4c,d, in HSV-2 (MOI = 3.0)-infected Vero cells, obvious syncytia with multinuclear cells were observed in the non-treated virus control group (HSV-2) at 7 h p.i. However, treatment with PSSD (50, 100 μg/mL) during 5–7 h p.i. markedly blocked the syncytium formation only with a limited number of small syncytia, suggesting that PSSD may block HSV-induced cell fusion (Figure 4c,d). Thus, PSSD may directly block HSV-induced membrane fusion through interaction with virus surface glycoproteins.

### 2.5. PSSD Exhibits In Vivo Antiviral Activity in HSV-2-Infected Mice

The anti-HSV-2 effects of PSSD were further tested in an HSV-2-induced murine genital herpes model [21]. As shown in Figure 5a,b, the mice body weight of the virus control group continuously reduced and sharply decreased after day 7 p.i. However, after PSSD treatment, the body weight of the mice slowly descended and eventually returned to the initial level (Figure 5a). In addition, compared with the untreated virus control group, the application of PSSD (2.5 or 5 mg·kg^−1^) gel significantly reduced the symptoms of genital herpes in mice (Figure 5b,d). The peak value and rising trend of symptom score in the PSSD-treated groups were markedly lower than those of virus control group and acyclovir (10 mg·kg^−1^)-treated group (Figure 5b).

To evaluate the inhibitory effect of PSSD on HSV-2 multiplication in vivo, the virus titer of HSV-2 in the mouse reproductive tract was determined by plaque assay. As shown in Figure 5c, compared with the non-treated virus control group, PSSD (2.5 or 5 mg·kg^−1^) treatment significantly reduced the titer of the mouse reproductive tract exfoliated virus by about 1.57 and 1.13 log_10_ (PFU/mL), respectively (*p* < 0.05), suggesting that vaginal gel therapy with PSSD can inhibit HSV-2 multiplication in mice. Acyclovir (10 mg·kg^−1^) treatment also showed a significant reduction of virus titers in mouse vagina (Figure 5c).

Furthermore, the inhibition of PSSD on genital herpes in mice was further evaluated by performing histopathology analysis of the genital tract tissues of mice. As shown in Figure 5e, the reproductive tract mucosa of mice in the virus control group showed thinning and damage. However, after treatment with PSSD (2.5 or 5 mg·kg^−1^) or acyclovir (10 mg·kg^−1^), the HSV-2-infected mice had relatively intact mucosa, accompanied by mucosal keratinization repair and mucosal thickening (Figure 5e). Thus, PSSD can also significantly attenuate the reproductive tract damage in HSV-2-infected mice.

## 3. Discussion

Marine-algae-derived polysaccharides such as carrageenans and alginate polysaccharides were reported to be able to prevent the binding of HSV to the cell surface [12]. In this study, we found that marine sulfated polysaccharide PSSD possessed significant anti-HSV-2 effects with low toxicity. PSSD can directly interact with HSV particles to inactivate the virus. PSSD may also interact with virus surface proteins to inhibit membrane fusion. In addition, the treatment of HSV-2-infected mice with PSSD gel significantly attenuated the symptoms of genital herpes and reduced the titer of exfoliative virus in the mice’s reproductive tracts, suggesting that PSSD has the potential to be developed into a novel anti-HSV-2 agent in the future.

The alginate-derived polymannuroguluronate sulfate (PMGS) was reported to exert the anti-HPV effects mainly through blocking the binding of HPV L1 protein to the host cell surface [14]. Similarly, the time-of-addition assay showed that PSSD treatment during adsorption also significantly inhibited the multiplication of HSV-2. However, pretreatment of HSV by PSSD also markedly reduced the expression of virus ICP27 in HSV-2-infected cells, suggesting that PSSD may have direct interaction with HSV-2 particles. Consistently, the results of the subsequent plaque reduction assay indicated that PSSD may be able to directly inactivate the virus particle. Thus, PSSD may possess a direct virucidal effect on HSV-2 particles, which is different from the nucleoside analogues such as acyclovir.

The envelope of HSV is a lipid bilayer containing 12 glycoproteins, among which four glycoproteins, gB, gD, gH, and gL, are necessary to induce cell membrane fusion, and are crucial for viruses to enter host cells [22]. In this study, the time course study within 8 h indicted that PSSD mainly inhibited the early steps (0–2 h p.i.) of the HSV-2 life cycle after adsorption, suggesting that PSSD may be able to block the entry process of HSV-2. Consistently, the subsequent membrane fusion inhibition assay showed that PSSD can truly block the virus-induced cell fusion process during 5–7 h p.i. Taken together, all of the results suggest that PSSD may also block the HSV-induced membrane fusion process through interaction with virus surface glycoproteins such as gB and gD. 

The HSV-2-induced murine genital herpes model has been established and used for studying the anti-HSV-2 effects of PSSD in vivo [21]. Herein, we found that vaginal gel treatment of PSSD significantly reduced the symptoms of genital herpes and inhibited HSV multiplication in the reproductive tract of mice. Moreover, the histopathology analysis indicated that PSSD treatment also significantly alleviated the reproductive tract damage of HSV-2-infected mice, superior to the effects of acyclovir, suggesting that PSSD also possessed marked anti-HSV-2 activities in vivo. Although, similarly to other sulfated polysaccharides, PSSD may hardly cross the different barriers of the body by oral administration, our studies showed that vaginal gel therapy of PSSD had remarkable anti-HSV-2 effects, which suggested that PSSD may be used for prevention and treatment of genital herpes by vaginal administration in the future.

## 4. Materials and Methods

### 4.1. Reagents, Cells and Viruses

PSSD was provided by the Marine Biomedical Research Institute of Qingdao (Qingdao, China). Acyclovir was purchased from Sigma Aldrich (St. Louis, MO, USA). HSV-2 strain 333 was obtained from the Wuhan Institute of Virology, Chinese Academy of Sciences. Vero cells were cultured in minimum essential medium (MEM) supplemented with 10% FBS (ExCell Bio, Shanghai, China), penicillin (100 U/mL), and streptomycin (100 μg/mL). HeLa cells were routinely cultured in Dulbecco’s modified Eagle’s medium (DMEM) supplemented with 10% fetal bovine serum (FBS) (ExCell Bio, China), penicillin (100 U/mL), and streptomycin (100 μg/mL) at 37 °C in 5% CO_2_. The anti-HSV-2 ICP27 antibody was purchased from Santa Cruz (Santa Cruz, CA, USA). 

### 4.2. Cytotoxicity Assay

The cytotoxicity of compounds was measured by the MTT (Sigma-Aldrich, St. Louis, MO, USA) assay [18]. Confluent Vero and HeLa cell cultures in 96-well plates were exposed to different concentrations of compounds in triplicate and incubated at 37 °C for 24 h. Next, 10 μL of PBS containing MTT (final concentration: 0.5 mg/mL) was added to each well. After 4 h incubation at 37 °C, the supernatant was removed and 200 μL of DMSO was added to each well to solubilize the formazan crystals. After vigorous shaking, absorbance values were measured in a microplate reader (Bio-Rad, Hercules, CA, USA) at 570 nm. The CC_50_ was calculated as the compound concentration necessary to reduce cell viability by 50%.

### 4.3. Cytopathic Effect (CPE) Inhibition Assay

The antiviral activity was evaluated by the CPE inhibition assay as described previously [19]. In brief, Vero cells were infected with HSV-2 (333 strain) at a multiplicity of infection (MOI) of 0.1, respectively, and then treated with indicated concentrations of PSSD (0.78125–200 μg/mL) in triplicate, after removal of virus inoculum. After 24 h incubation, the cells were fixed with 4% formaldehyde for 20 min at room temperature (RT). After removal of the formaldehyde, the cells were stained with 0.1% (*w*/*v*) crystal violet for 30 min at 37 °C. The plates were washed and dried, and the intensity of crystal violet staining for each well was measured at 570 nm. 

### 4.4. Time-of-Addition Assay

Vero cells were infected with HSV-2 (MOI = 1.0) under four different treatment conditions: (i) Pretreatment of virus: PSSD (100 µg/mL)-pretreated HSV-2 was added to Vero cells and incubated at 37 °C for 1 h. Then after adsorption, the virus inoculum containing PSSD was removed and the cells were overlaid with compound-free media. (ii) Pretreatment of cells: Vero cells were pretreated with 100 µg/mL of PSSD before HSV-2 infection. (iii) Adsorption: Vero cells were infected in media containing PSSD (100 μg/mL) at 4 °C for 1 h. After that, the virus inoculum was removed and the compound-free media were added into cells. (iv) Post-adsorption: after removal of unabsorbed virus, PSSD (100 µg/mL) was added to the cells. At 24 h p.i., virus yields were determined by plaque assay or western blot assay. 

Moreover, another time-dependent study was also performed to explore which viral stage after adsorption is inhibited by PSSD as described previously [19]. Briefly, HSV-2 (MOI = 1.0)—infected Vero cells were treated with 100 μg/mL of PSSD for different time intervals (0–2 h p.i., 2–4 h p.i., 4–6 h p.i., 6–8 h p.i.), after which (at 24 h p.i.) the virus yields were determined via western blot assay of virus ICP27 protein. The relative densities of protein bands were determined by Image J V.1.33 u (NIH, Bethesda, MD, USA).

### 4.5. Plaque Reduction Assay

Approximately 100–200 PFU per well of HSV-2 was pre-incubated with PSSD (12.5–100 μg/mL) for 1 h at 37 °C before infection, respectively. Then the virus–PSSD mixture was transferred to Vero cell monolayers in 12-well plates, and incubated at 4 °C for 1 h. After removal of the inoculum, cells were washed and overlaid with 2 mL of agar overlay media containing 1.5% agarose, 100 U/mL of penicillin, and 100 μg/mL of streptomycin. After incubation for 3 days at 37 °C in a humidified atmosphere of 5% CO_2_, cells were fixed with 0.05% glutaraldehyde, followed by staining with 1% crystal violet in 20% ethanol for plaque counting.

### 4.6. Western Blot Assay

After drug treatment, the cell lysate was separated by SDS-PAGE and transferred to nitrocellulose membrane. After being blocked in Tris-buffered saline (TBS) containing 0.1% Tween 20 (*v*/*v*) and 5% BSA (*w*/*v*) at RT for 2 h, the membranes were rinsed and incubated at 4 °C overnight with primary antibodies against virus ICP27 proteins or cellular anti-β-actin antibodies (Cell Signaling Technology, Danvers, MA, USA) as a control. The membranes were washed and incubated with AP-labeled secondary antibody (1:5000 dilutions) at RT for 2 h. The protein bands were then visualized by incubating with the developing solution (p-nitro blue tetrazolium chloride (NBT) and 5-bromo-4-chloro-3-indolyl phosphate toluidine (BCIP)) at RT for 30 min. The relative densities of proteins were all determined by using Image J V.1.33 u (NIH, Bethesda, MD, USA).

### 4.7. Syncytium Formation Inhibition Assay

Syncytium assays were performed using the method described previously with some modifications [23]. Briefly, Vero cells were first infected with HSV-2 (MOI = 3.0) for 2 h at 4 °C before being washed and incubated at 37 °C for 5 h. Then PSSD (50, 100 μg/mL) was added to cells and incubated at 37 °C for 2 h. After that, cells were fixed and then stained with haematoxylin–eosin solution (Beyotime, Nantong, China) before observation. The inhibition of syncytium formation was observed using an optical microscope. To quantify the degree of syncytium inhibition, the relative syncytium areas of different fields were measured using ImageJ (NIH) V.1.33u. The percentage of syncytium formation was determined relative to the non-treated virus control group (HSV-2).

### 4.8. Animal Experiments

All animal experiments were performed in accordance with the National Institutes of Health guide for the care and use of laboratory animals and approved by the Institutional Animal Care and Use Committee at Ocean University of China (OUC-SMP-2022-12-10). Eight-week-old and sexual maturity female BALB/c mice (average weight, 18.0 ± 2.0 g) were purchased from Beijing Vital River Laboratory Animal Technology Co., Ltd. (Beijing, China) and raised in a pathogen-free environment (23 ± 2 °C and 55 ± 5% humidity). Mice were randomly divided into five experimental groups (10 mice each). Three days before modeling, each mouse was subcutaneously injected with 30 mg/mL medroxyprogesterone 100 μL [24]. On the day of modeling, the mice were anesthetized by intraperitoneal injection of 30% galactose according to the body weight ratio of the mice, and the cell brush was used to rub the vagina of the mice back and forth. After the vaginal bleeding of the mice was fully opened, 20 μL compound gel (or blank gel) and 20 μL HSV-2 (2 × 10^5^ PFU/mL) virus solution were successively injected into the reproductive tract of the mice using a pre-sterilized pipette. Four hours after inoculation, mice were treated with gel application of acyclovir (10 mg·kg^−1^), PSSD (2.5 or 5 mg·kg^−1^) or placebo twice daily for five consecutive days in the morning and evening, respectively.

Each day, mice were weighed and monitored for signs of illness for 13 days, and those suffering a severe infection or having lost >20% of their original body weight were euthanized. To determine the viral titer in the organs, the vaginal lavage fluid of mice was taken on the third day after inoculation to carry out the plaque assay. Histopathological analysis was performed using H&E staining on vagina samples collected on day 9 [25].

### 4.9. Statistical Analysis

All data are representative of at least three independent experiments. Data are presented as means ± standard deviations (S.D.). Statistical significance was analyzed using GraphPad Prism 8.0 software. Comparison between groups was performed by using One-Way ANOVA analysis followed by Tukey’s test, with *p* values < 0.05 considered significant.

## 5. Conclusions

In summary, PSSD possessed anti-HSV-2 activities both in vitro and in vivo with low toxicity. PSSD may be able to directly inactivate the virus particle and interact with virus surface proteins to the block membrane fusion process. Further studies of the anti-HSV effects of PSSD against clinical strains, especially the acyclovir-resistant strains, will be required to advance it for drug development. In a word, marine polysaccharide PSSD has the potential to be developed into a novel anti-HSV-2 agent for prophylaxis and therapy of genital herpes in the future.

## Figures and Tables

**Figure 1 marinedrugs-21-00364-f001:**
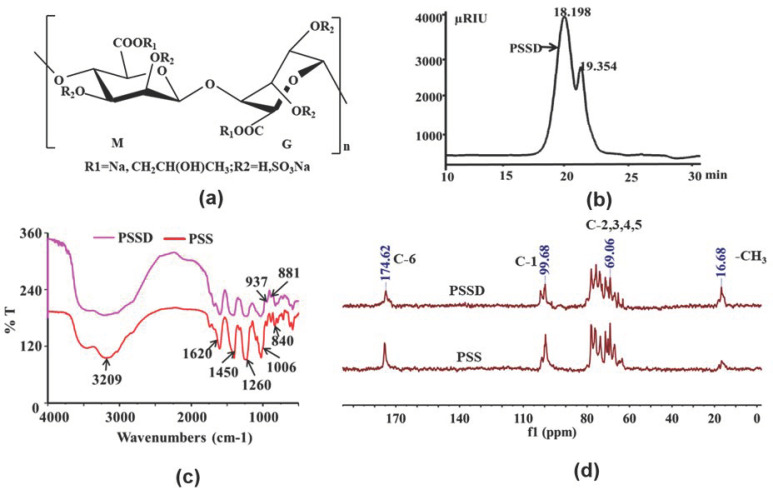
Structure characterization of marine polysaccharide PSSD. (**a**) Schematic illustration of PSSD structure. (**b**) The HPGPC chromatogram of PSSD. (**c**) The FT-IR spectra of PSS and PSSD. (**d**) The ^13^C NMR spectra of PSS and PSSD.

**Figure 2 marinedrugs-21-00364-f002:**
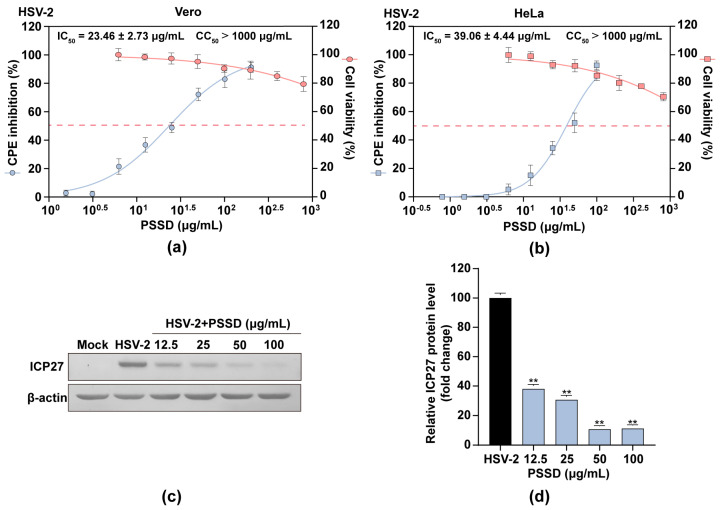
Inhibition effects of PSSD against HSV-2 infection in vitro. (**a**,**b**) The cytotoxicity and anti-HSV-2 effects of PSSD in Vero and HeLa cells. Values are means ± S.D. (*n* = 3). (**c**,**d**) The effect of PSSD on HSV-2 multiplication was also evaluated by western blot assay of virus ICP27 protein in Vero cells (**c**). Quantification of immunoblot for the ratio of ICP27 to β-actin was also shown (**d**). Values are means ± S.D. (*n* = 3). ** *p* < 0.01 vs. virus control group (HSV-2).

**Figure 3 marinedrugs-21-00364-f003:**
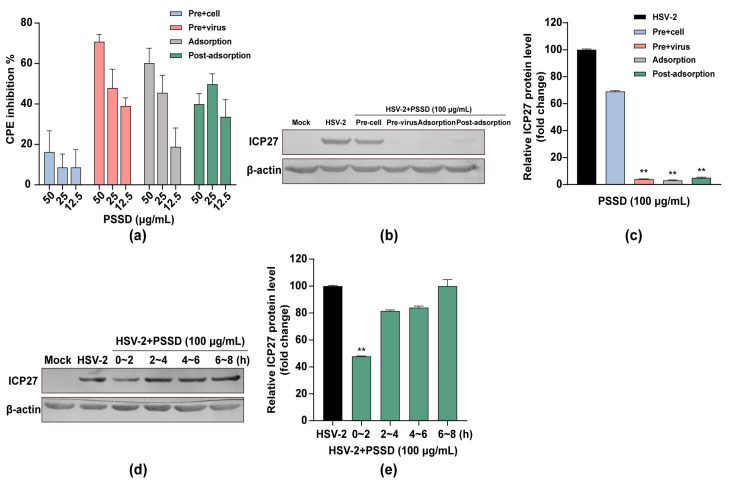
Influence of different treatment conditions of PSSD on HSV-2 infection. (**a**) Vero cells were infected with HSV-2 (MOI = 0.1) under four treatment conditions of PSSD (12.5–50 μg/mL) and the antiviral activity was determined by CPE inhibition assay at 24 h p.i. Values are means ± S.D. (*n* = 3). (**b**,**c**) Vero cells were infected with HSV-2 (MOI = 1.0) under four treatment conditions of PSSD (100 μg/mL) and the virus yields were determined via western blot assay of virus ICP27 protein (**b**). Quantification of immunoblot for the ratio of ICP27 to β-actin was also shown (**c**). Values are means ± S.D. (*n* = 3). ** *p* < 0.01 vs. virus control group (HSV-2). (**d**,**e**) HSV-2 (MOI = 1.0)-infected Vero cells were treated with 100 μg/mL of PSSD for different time intervals, after which (at 24 h p.i.) the virus yields were determined via western blot assay of virus ICP27 protein (**d**). Quantification of immunoblot for the ratio of ICP27 to β-actin was also shown (**e**). Values are means ± S.D. (*n* = 3). ** *p* < 0.01 vs. virus control group (HSV-2).

**Figure 4 marinedrugs-21-00364-f004:**
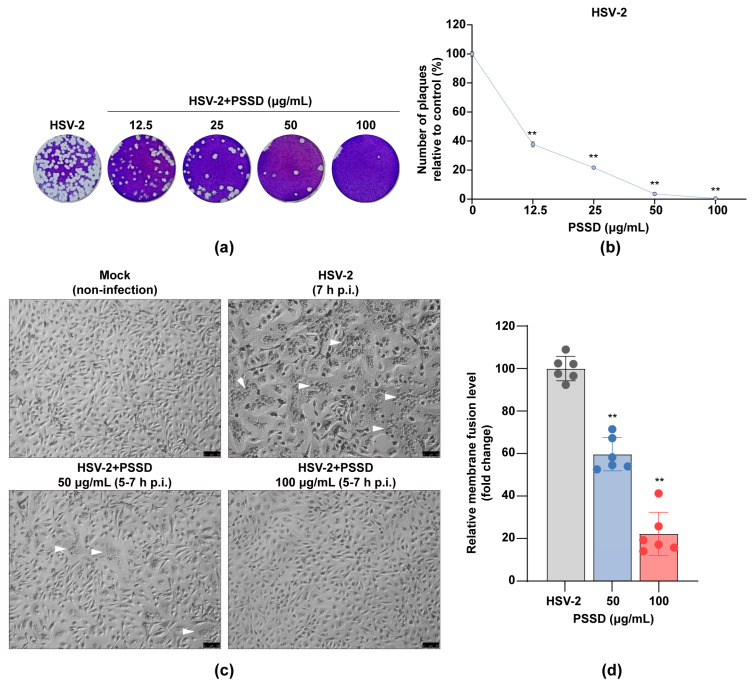
PSSD possesses direct actions on HSV particles to block membrane fusion. (**a**) The direct actions of PSSD on HSV-2 particles were evaluated by plaque reduction assay in Vero cells. (**b**) Plots quantifying the plaque numbers from plaque reduction assay (**a**) was shown, and the percent level was determined relative to the virus control. Values are means ± S.D. (*n* = 3). ** *p* < 0.01 vs. virus control group (0 μg/mL). (**c**) The inhibition of PSSD on HSV-2-induced membrane fusion was evaluated by syncytium inhibition assay in Vero cells. Bar represents 50 μm. (**d**) Plots quantifying syncytium formation in HSV-2-infected cells with different treatments. Values are means ± S.D. (*n* = 3). ** *p* < 0.01 vs. virus control group (HSV-2). White triangle indicates the position of membrane fusion.

**Figure 5 marinedrugs-21-00364-f005:**
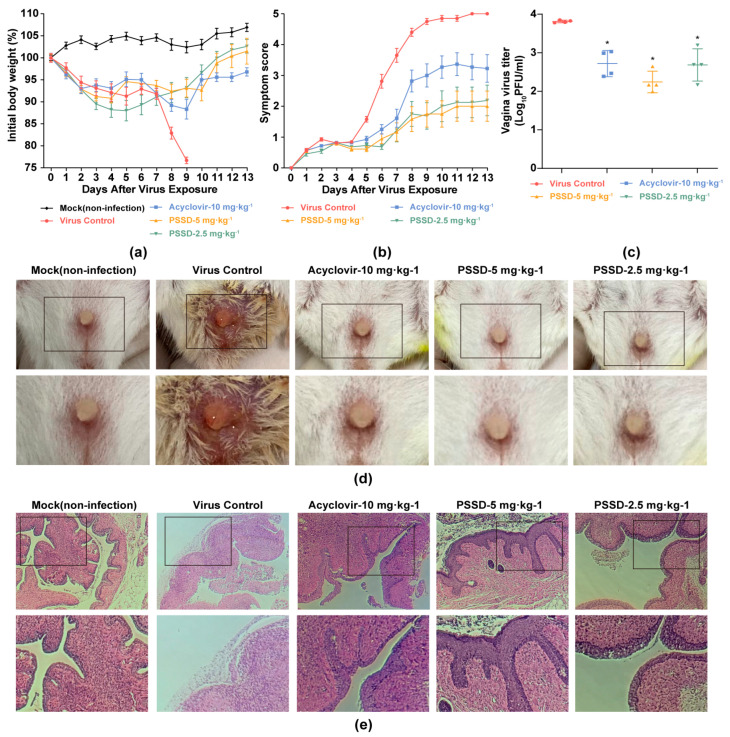
The anti-HSV-2 activities of PSSD in vivo. (**a**) Body weight. The body weights of 10 mice in each group were monitored daily for 13 days and are expressed as a percentage of the initial value. (**b**) Symptom score. The extent of disease progression in HSV-2-infected mice was determined by daily pathology scoring (0–5). (**c**) Viral titers in vaginal were evaluated by plaque assay. Values are means ± S.D. (*n* = 4). * *p* < 0.05 vs. virus control group. (**d**) Pictures displaying the ulceration and inflammation of vaginal tissues of HSV-2-infected mice on day 9. (**e**) Histopathology analysis of vaginal tissues on day 3 p.i. (×10 and ×20).

## Data Availability

The data presented in this study are available on request from the corresponding author.

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
