# Peer review of "Inhibition Effects and Mechanisms of Marine Polysaccharide PSSD against Herpes Simplex Virus Type 2"

_marinedrugs, 2023, doi:10.3390/md21060364_

Round 1
Reviewer 1 Report
Yan et al. performed an interesting study that showed a particular marine polysaccharide, namely PSSD, had antiviral activity towards herpes simplex virus (HSV) both in vitro and in vivo. Of note, the in vivo activity of PSSD is somewhat better than that of acyclovir, an existing anti-HSV drug. However, there are a few minor issues.
First, the authors should clarify what “PSSD” stands for.
Second, the 13C NMR spectrum in Figure 1d. was not properly phased.
Third, and more importantly, the authors should elaborate that PSSD is a mixture, as shown in Figure 1, with molecular components of different substituents and degrees of polymerization. Hence, a question regarding PSSD purity follows: how exactly was its purity determined? The authors mentioned PSSD (with purity > 98%) was used for this study, yet we don’t know its exact composition.
English language mostly fine. Minor edits are required.
Author Response
Reviewing: 1
Comments to the Author:
Yan et al. performed an interesting study that showed a particular marine polysaccharide, namely PSSD, had antiviral activity towards herpes simplex virus (HSV) both in vitro and in vivo. Of note, the in vivo activity of PSSD is somewhat better than that of acyclovir, an existing anti-HSV drug. However, there are a few minor issues.
Authors: Thanks a lot for your comments and advice. We have revised the manuscript accordingly as you suggested. Here are our point-by-point responses.
- The authors should clarify what “PSSD” stands for.
Thanks. PSS (Propylene glycol alginate sodium sulfate) is a sulfated alginate polysaccharide which has been used for preventing and treating hyperlipidemia and ischemic cardio-cerebrovascular diseases in China for 30 years. Recently, our group graded the alginate polysaccharide PSS and obtained four components with different molecular weights (29.6 kDa, 15.2 kDa, 9.4 kDa, 6.7 kDa). Through activity screening, it was found that the component with 6.7 kDa had marked anti-HSV activity in vitro, thus it was named “PSS derivative (PSSD)” and used for the subsequent studies. We also added some contents about PSSD in the Introduction section of the revised manuscript (Page 2, Lines 57-63 in the revised paper).
- The 13C NMR spectrum in Figure 1d. was not properly phased.
Agree. In the revised paper, we reconfirmed the structures of PSS and PSSD with 13C NMR analysis and found that PSSD and PSS had similar structural features, and they were only different in their molecular weights. We had made some revision about the interpretation about the 13C NMR analysis (Page 2, Lines 92-100 in the revised paper). The 13C NMR spectrum in Figure 1d had been replaced and the figure legend of Figure 1 was also changed in the revised Figure legends section (Page 4, Lines 112-114 in the revised paper).
- The authors should elaborate that PSSD is a mixture, as shown in Figure 1, with molecular components of different substituents and degrees of polymerization. Hence, a question regarding PSSD purity follows: how exactly was its purity determined? The authors mentioned PSSD (with purity > 98%) was used for this study, yet we don’t know its exact composition.
Right. PSS and PSSD are mixtures of polysaccharides with different substituents and degrees of polymerization. In our preliminary studies, we graded the alginate polysaccharide PSS and obtained four components with different molecular weights (29.6 kDa, 15.2 kDa, 9.4 kDa, 6.7 kDa). PSSD is the component of PSS with the average molecular weight of 6.7 kDa. To determine the relative purity of PSSD, we used the mass balance method to calibrate the content of PSSD. Content% = (1- Moisture% - Impurities% - Volatile Matter%) × 100%; The moisture content of PSSD was calculated to be 1.21% using the drying method; The impurities were mainly free ash, and the free ash content was determined using ion chromatography, and it was found that PSSD contains 0.03% Cl- and 0.02% SO42-; The volatile substance was residual formamide solvent, and its content is determined by liquid chromatography, with a content less than 0.022%; Thus, Content% = (1-1.21% -0.03% -0.02%-0.022%) × 100% = 98.7%. In the revised manuscript, we had added some information about the origin and structure of PSSD in the revised Introduction and Results section (Page 2, Lines 57-63, Lines 78-80 in the revised paper).

Reviewer 2 Report
The title of the manuscript is good. English language has good quality. Figures needs some changes. Various sections of the manuscript need some changes.
1. Please rewrite the part "Introduction" according to order below:
+ Information about Herpes simplex virus
+ The importance of treating infection caused by Herpes simplex virus
+ the potential of marine environment in the treatment of infection caused by Herpes simplex virus
+ the potential of marine polysaccharides in the treatment of infection caused by Herpes simplex virus
+ the potential of PSSD in the treatment of infection caused by Herpes simplex virus
2. Line 65-69 in page 2
Please omit this part from the section "Results" this part belongs to other sections
of the manuscript
3. Page 2, line 81-83
This part should be deleted from the section "Results" because it belongs to the part "material and methods"
4. About the part "Results"
First: All over this part there are some sentences with reference, please omit all of them
Second: in this part, the authors should only write about their findings. Other data about other parts like any comoarisons with the results of other studies, any information about "material and methods" and any other extra data that belongs to any section other than "results" should be deleted
Third: the purpose of performing any tests should be mentioned in the section " material and methods", not in "Results". Please consider this note
5. The title of each figure contains information about other sections of manuscript (specially material and methods)
Please reform the titles of the fivures of manuscript (the title of each figure should only explain about that figure, not other parts of manuscript)
6. About the part "Discussion"
Please rewrite this part according to notes below:
First: categorize all of your results based on their importance (from the most important one to the least important)
Second: after that, turn each one of your results into some subheadings
Third: after that, discuss about them one by one
Forth: make comparisons between your results and the results of other similar and
relevant surveys
7. Please check and adjust the "Reference list" based on the regulations of reference list of journal. (Titles, doi, the name of journal and ... )
Author Response
Reviewing: 2
Comments to the Author
The title of the manuscript is good. English language has good quality. Figures needs some changes. Various sections of the manuscript need some changes.
Authors: Thank you very much for your comments and advice. We have revised the paper according to your suggestions. Here, we are willing to answer your questions point-to-point as follows.
- Please rewrite the part "Introduction" according to order below: + Information about Herpes simplex virus; + The importance of treating infection caused by Herpes simplex virus; + the potential of marine environment in the treatment of infection caused by Herpes simplex virus; + the potential of marine polysaccharides in the treatment of infection caused by Herpes simplex virus; + the potential of PSSD in the treatment of infection caused by Herpes simplex virus.
Thanks. In the revised paper, we had rewritten the part "Introduction" follow your suggestions. We had added some contents about the potential of marine environment and PSSD in the treatment of infection caused by HSV (Page 1, Lines 42-45, and Page 2, Lines 57-65 in the revised paper). We also added some information about the origin of PSSD in the revised Introduction section (Page 2, Lines 57-62 in the revised paper).
- Line 65-69 in page 2: Please omit this part from the section "Results" this part belongs to other sections of the manuscript.
Agree. In the revised paper, we had omitted this part from the section "Results", and added some information about PSS and PSSD into the revised Introduction section (Page 2, Lines 57-62 in the revised paper).
- Page 2, line 81-83: This part should be deleted from the section "Results" because it belongs to the part "material and methods".
Right. In the revised paper, we had deleted this part from the section "Results". We also added some content about the 13C NMR analysis of both PSS and PSSD in the revised Results section (Page 2, Lines 92-98 in the revised paper).
- About the part "Results": 1) All over this part there are some sentences with reference, please omit all of them. 2) in this part, the authors should only write about their findings. Other data about other parts like any comoarisons with the results of other studies, any information about "material and methods" and any other extra data that belongs to any section other than "results" should be deleted. 3) the purpose of performing any tests should be mentioned in the section " material and methods", not in "Results". Please consider this note.
Thanks. In the revised paper, we had rewritten some contents in Results section and removed the sentences about the detailed methods, extra data, and the purpose of experiments from Results section (Page 4, Lines 123-126; Page 5, Lines 144-149, Lines 160-164; Page 6, Lines 190-191; Page 7, Lines 223-225; Page 8, Lines 241-243 in the revised paper). We had removed some information about the experiment conditions to the revised Materials and Methods section.
- The title of each figure contains information about other sections of manuscript (specially material and methods). Please reform the titles of the figures of manuscript (the title of each figure should only explain about that figure, not other parts of manuscript).
Agree. In the revised paper, we had reformed the titles of the figures in the revised Figure legends section, and removed the additional information such as the detailed methods from all of the Figure legends (Page 5, Lines 135-140; Page 6, Lines 172-177; Page 7, Lines 206-220; Page 8, Lines 250-260 in the revised paper).
- About the part "Discussion"
Please rewrite this part according to notes below: First: categorize all of your results based on their importance (from the most important one to the least important); Second: after that, turn each one of your results into some subheadings; Third: after that, discuss about them one by one; Forth: make comparisons between your results and the results of other similar and relevant surveys.
Thanks. In the revised paper, we had rewritten the Discussion section according to notes as you suggested. We also added some discussion about the inhibition effect of PSSD on HSV-2 induced membrane fusion process in the revised Discussion section (Page 9, Lines 264-266, Lines 274-294 in the revised paper).
- Please check and adjust the "Reference list" based on the regulations of reference list of journal. (Titles, doi, the name of journal and ...).
Agree. In the revised paper, we had checked and adjusted the "Reference list" according to the format requirements of reference list of this journal. We also added two other references into the revised Reference section (Lines 444-504 in the revised paper).

Round 2
Reviewer 2 Report
I do not have more comment.